# Anti-Obesity Effects of Isorhamnetin and Isorhamnetin Conjugates

**DOI:** 10.3390/ijms24010299

**Published:** 2022-12-24

**Authors:** Maitane González-Arceo, Iván Gomez-Lopez, Helen Carr-Ugarte, Itziar Eseberri, Marcela González, M. Pilar Cano, María P. Portillo, Saioa Gómez-Zorita

**Affiliations:** 1Nutrition and Obesity Group, Department of Nutrition and Food Science, Lucio Lascaray Research Institute, University of the Basque Country (UPV/EHU), 01006 Vitoria-Gasteiz, Spain; 2CIBERobn Physiopathology of Obesity and Nutrition, Institute of Health Carlos III, 28029 Madrid, Spain; 3Laboratory of Phytochemistry and Plant Food Functionality, Biotechnology and Food Microbiology Department, Institute of Food Science Research (CIAL) (CSIC-UAM), 28049 Madrid, Spain; 4BIOARABA Health Research Institute, 01006 Vitoria-Gasteiz, Spain; 5Nutrition and Food Science Department, Faculty of Biochemistry and Biological Sciences, National University of Litoral and National Scientific and Technical Research Council (CONICET), Santa Fe 3000, Argentina

**Keywords:** isorhamnetin, isorhamnetin glucosides, obesity, adipose tissue, adipocytes, animal models

## Abstract

Isorhamnetin is a plant-derived secondary metabolite which belongs to the family of flavonoids. This review summarises the main outcomes described in the literature to date, regarding the effects of isorhamnetin on obesity from in vitro and in vivo studies. The studies carried out in pre-adipocytes show that isorhamnetin is able to reduce adipogenesis at 10 μM or higher doses and that these effects are mediated by Pparγ and by Wnt signalling pathway. Very few studies addressed in rodents are available so far. It seems that treatment periods longer than two weeks are needed by isorhamnetin and its glycosides to be effective as anti-obesity agents. Nevertheless, improvements in glycaemic control can be observed even in short treatments. Regarding the underlying mechanisms of action, although some contradictory results have been found, reductions in *de novo* lipogenesis and fatty acid uptake could be proposed. Further research is needed to increase the scientific evidence referring to this topic; studies in animal models are essential, as well as randomised clinical trials to determine whether the positive results observed in animals could also be found in humans, in order to determine if isorhamnetin and its glycosides can represent a real tool against obesity.

## 1. Introduction

Isorhamnetin is a plant-derived secondary metabolite which belongs to the family of flavonoids, and more specifically, to the group of flavonols. Flavonoids comprise a large group of bioactive phytochemicals consisting of two phenolic benzene rings and a heterocyclic ring (Figure 1). Isorhamnetin is known for being a 3′-*O*-methylated gut metabolite of quercetin, but it can also be found in several plant species, such as *Hippophae rhamnoides*, *Ginkgo biloba* or *Opuntia stricta* var. *dilleni*, which have traditionally been used as medicinal plants in several cultures [1,2]. Other primary sources of isorhamnetin that are part of the human diet are onions, almonds or various berries [2,3] (Table 1). Additionally, isorhamnetin conjugated forms have also been identified in many foods such as pears, grapes, apples, berries, almonds or cherries and plant species such as *Opuntia stricta* var. *dilleni* or *Hippophae rhamnoides*, among others [2,4].

Flavonoids have displayed many beneficial health effects in the prevention and/or treatment of cardiovascular diseases, cancer, type 2 diabetes mellitus, age-related diseases and obesity, among others [1,11,12,13]. More specifically, isorhamnetin is known to exert different biological activities including anti-oxidant, anti-proliferative, anti-inflammatory, anti-diabetic and anti-obesity actions [14].

Obesity is a major public health concern, not only due to its direct effect on body weight, but also because it is an important risk factor for the development of other metabolic disorders such as type 2 diabetes mellitus, fatty liver disease, cardiovascular diseases, or some types of cancers (i.e., breast or colon). Strategies that manage obesity aimed at reducing calorie intake and increasing energy expenditure often fail in the long term [15]. In this context, it is necessary to look for novel therapeutic strategies with anti-obesity activity to complement the more traditional tools.

The present narrative review summarises the main outcomes described in the literature so far in relation to the effects of isorhamnetin on obesity from in vitro and in vivo studies, providing an in-depth description of the mechanisms underlying its potential effects on this pathology.

## 2. Literature Search

A search was conducted in the PubMed database in September 2022. The terms used in title/abstract were isorhamnetin and obesity or adipocytes or adipose tissue. There were no limitations placed on the literature search pertaining to the publication date of the studies. However, reviews were excluded.

## 3. Studies in Cultured Adipocytes

Several studies have evaluated the in vitro effects of isorhamnetin and isorhamnetin conjugates, mainly in both maturing pre-adipocytes and mature adipocytes from different cell lines, the murine 3T3-L1 cell line being the most used one (Table 2).

Adipogenesis is a complex process that involves the differentiation of multipotent adipose-derived stem cells into pre-adipocytes and the conversion of these cells into mature adipocytes. It plays a key role in the development of obesity, mainly in young subjects. This process involves numerous transcription factors that coordinate the expression of many proteins responsible for setting-up the mature adipocyte phenotype [25,26], the peroxisome proliferator-activated receptor γ (PPARγ) being the master regulator [27]. 

In order to analyse the effect of isorhamnetin on this process, Lee et al. (2009) incubated pre-adipocyes with 1, 10, 25 and 50 μM of this molecule for nine days [16]. The highest doses (25 and 50 μM) significantly reduced triglyceride content, but the lowest doses did not prevent lipid accumulation. To characterise the mechanism underlying this effect the dose of 50 μM was selected. The expression of *Pparγ*, CCAAT-enhancer-binding protein α (*C/ebpα*), peroxisome proliferator–activated receptor-γ coactivator *1* (*Pgc-1*) and the target genes of PPARγ, lipoprotein lipase (*Lpl*), cluster of differentiation 36 (*Cd36*), adipose fatty acid-binding protein 2 (*Ap2*) and liver X receptor α (*Lxrα*) were reduced. By contrast, CCAAT-enhancer-binding protein β and δ (*C/ebpβ* and *C/ebpδ*) and *Krox 20* (also known as early growth response 2 [Egr2]) were not downregulated by the treatment. Gene expression of adiponectin, a marker of mature adipocytes, and its secretion were reduced. Moreover, the activity of glycerol-3-phosphate dehydrogenase (GPDH), an enzyme which plays an important role in the conversion of glycerol into triglycerides, was reduced. These results suggest that isorhamnetin inhibits adipocyte differentiation via PPARγ and C/EBPα. 

The same group subsequently analysed the effect of the highest dose of isorhamnetin used in the previous study (50 μM) on the activation of the Wnt/β-catenin pathway, which inhibits adipogenesis by stabilizing β-catenin [28,29,30]. The incubation was carried out during the first six days of the differentiation period. Cells treated with isorhamnetin showed a lower lipid accumulation than the control cells. Moreover, they exhibited increased Wnt/β-catenin activity, as well as increased content of β-catenin in its nuclear localization. These results suggest the involvement of the Wnt/β-catenin pathway on the anti-adipogenic effect of isorhamnetin [27].

Using the same cell line, Zhang et al. (2016) observed decreased cell triglyceride content with the three doses used in their experiment (10, 25 and 50 μM) [19]; this effect followed a dose-dependent pattern. Interestingly, in this study, the authors would elucidate the effect of isorhamnetin on adipocyte differentiation at the different stages of the adipogenesis. To do this, pre-adipocytes were treated with a dose of 50 μM at days zero, three and six. The cells triglyceride content was lower when isorhamnetin was added to the incubation medium at day three or later, indicating that isorhamnetin inhibits late stages of adipocyte differentiation. The gene expression analysis revealed that, when cells were treated at day zero, mRNA levels of *C/ebpα*, *Ap2* and *Cd36* decreased, although *Pparγ* or *C/ebpβ* remained unchanged. In contrast, when cells were treated at the late stage of the differentiation period (day six), a reduction in *Pparγ*, *Ap2*, *Cd36*, fatty acid synthase (*Fasn*), acetyl coenzyme A carboxylase (*Acc*), uncoupling protein 2 (*Ucp2*), and *Lpl* mRNA levels was observed. In conclusion, isorhamnetin inhibited adipogenesis, attenuating *Pparγ* gene expression and its target genes under these experimental conditions.

Lee and Kim (2018) carried out a study by using doses lower than those used in the previous studies to analyse cell viability [20]. They incubated 3T3-L1 pre-adipocytes with isorhamnetin (0.1, 0.5, 1, 10, 20, 50 μM) for one, two, five or seven days and they observed that this parameter was significantly decreased only at 50 μM after seven days of treatment; in the range 0.1–20 μM there was no cytotoxic effect. Then, the authors used a concentration range below 20 μM to study the effect of isorhamnetin on lipid accumulation. For this purpose, both triglyceride content and GPDH activity were measured after the addition of 1, 10 or 20 μM of isorhamnetin to the incubation media from day two to day nine, and a significant reduction was observed in cells treated with the highest dose (20 μM). Regarding adipogenic gene expression, that of *Pparγ* and *Ap2* was significantly decreased. Altogether, the results show that at 20 μM, but not at lower doses, isorhamnetin inhibited adipogenesis.

In the study reported by Ganbold et al. (2019) the effects of 10 or 20 μM of isorhamnetin in murine 3T3-L1 pre-adipocytes were analysed for a longer incubation period (14 days) [21]. Both doses significantly decreased the formation of lipid droplets when compared with the control cells, indicating adipogenesis inhibition. These authors did not study the underlying mechanisms involved in the anti-adipogenic effect of the molecule.

Lastly, in a study addressed by our research group, where 0.1, 1 or 10 μM of isorhamnetin were added to 3T3-L1 pre-adipocyte for eight days, only the dose of 10 μM was able to significantly reduce triglyceride content [22]. The treatment with 10 μM did not modify the expression of *C/ebpβ,* sterol regulatory element-binding factor 1 (*Srebf1*) and *C/ebpα*; however, a tendency towards reduced mRNA levels was observed in the master regulator of adipogenesis *Pparγ*. In addition, genes related to the apoptotic pathway, *Trp53, Cas3* and *Bcl2,* were measured. It was shown that *Trp53* gene expression was increased and *Bcl2* gene expression was reduced, although no significant changes were observed in *Cas3* mRNA levels, suggesting that isorhamnetin was not able to completely activate the apoptotic pathway.

The effect of isorhamnetin on adipogenesis has also been studied in human cells. Lee et al. (2010) studied the influence of this compound on the adipogenic differentiation of human mesenchymal stem cells (hMSCs) derived from adipose tissue and the potential implication of the Wnt pathway [17]. For this purpose, isorhamnetin at 1, 10, 25 or 50 μM was added to the cell incubation medium for 21 days. At 1 μM, this molecule did not reduce cell triglyceride content, but at higher doses significant reductions were observed, suggesting a dose-dependent pattern. Bearing that in mind, a dose of 25 μM was selected to study the mechanism of action. Wnt signalling was controlled by soluble extracellular antagonists, including secreted frizzled-related proteins (SFRP), Wnt inhibitory factor-1 (WIF-1), cerberus and dickkopf (DKK), which promote adipogenesis through the degradation of β-catenin [31]. In this study, hMSC treated with isorhamnetin showed a lower *Sfrp1* and *Dkk1* gene expression than the control cells. However, no differences in *Sfrp2*, *Sfrp3*, *Sfrp4* and *Dkk3* were observed. These results suggest that sFRP1 and Dkk1 are involved in the anti-adipogenic effect of isorhamnetin in hAMSCs. Moreover, it has been proposed that the inhibition of the Wnt pathway by the mentioned antagonists is mediated by FZD receptors (SFRP) and LDL Receptor Related Protein (LRP) receptors (DKKs) [31,32]. Thus, in the present study, the authors measured the effects of isorhamnetin in the gene expression of these receptors and they found a downregulation of *Lrp5*, *Lrp6*, *Fzd4*, Fzd6 and *Fzd7* mRNA. Regarding β-catenin, no changes in its mRNA levels were observed, but glycogen synthase kinase-3 GSK β (GSK-3β), involved in β-catenin degradation, was inhibited. These effects resulted in increased nuclear levels and total protein levels of β-catenin in isorhamnetin-treated cells. Altogether, these findings show that isorhamnetin inhibits adipogenic differentiation of hMSCs by the stabilization of β-catenin.

Concerning mature adipocytes, a small number of studies have been addressed. Lee et al. (2009) studied the effect of isorhamnetin on the expression of several genes related to lipid metabolism and inflammation [16]. The 3T3-L1 adipocytes were treated after nine days of differentiation with 25 or 50 μM of the phenolic compound for 20 h. After that period, they observed that, whereas *Pparγ* gene expression was not modified by the treatment, interleukin-6 (*Il-6)*, monocyte chemoattractant protein-1 (*Mcp*-1) and plasminogen activator inhibitor-1 (*Pai-1)* mRNA levels were downregulated in a dose–response manner. 

In the study previously described in this review and addressed by our research group, 3T3-L1 mature adipocytes were incubated with 0.1, 1 or 10 μM of isorhamnetin for 24 h on day 12 after the induction of differentiation [22]. None of the tested doses promoted any changes in triglyceride content.

The effect of isorhamnetin conjugated with other molecules (Figure 2) during adipogenesis has also been studied, although to a lesser extent. 

In this line, Kong and Seo (2012) evaluated the effect of isorhamnetin 3-*O*-β-d-glucopyranoside on adipogenic differentiation in 3T3-L1 cells [23]. Pre-adipocytes were treated with 1, 10 or 20 μM of this compound during the six days of the differentiation period. Cell viability was evaluated at the end of the experimental period and no cytotoxic effect was observed at selected doses. Concerning the effect on intracellular lipid accumulation during the differentiation, it was shown that this phenolic compound reduced the lipid droplets in adipocytes at all used concentrations. Regarding the mechanism of action, isorhamnetin decreased the protein expression of PPARγ, SREBP1 and C/EBP*α*. Moreover, it also downregulated the protein expression of mature adipocytes markers, such as fatty acid synthase (FAS), GLUT4, RXRα, and leptin. AMP-activated protein kinase (AMPK) activity was also increased by isorhamnetin. All in all, the results of this study suggested that isorhamnetin 3-*O*-β-d-glucopyranoside shows anti-adipogenic activity mediated via AMPK activation. 

In the study reported by Im et al. (2017), 3T3-L1 pre-adipocytes were differentiated and treated from day 4 of differentiation with 5, 10 or 20 μM of isorhamnetin-3-*O*-d-glucuronide, and triglyceride content was measured on day eight [24]. The compound inhibited lipid accumulation for the whole range of concentrations in a dose-dependent manner. The mechanisms of action were not studied by the authors.

### 3.1. Summary

The studies carried out in 3T3-L1 pre-adipocytes reveal that isorhamnetin is able to reduce adipogenesis. In these studies, the doses used ranged from 0.1 to 50 μM, but only 10 μM and higher doses were effective. The vast majority of the studies reported did not analyse whether isorhamnetin affected the whole adipogenic process or just one phase. By incubating pre-adipocytes during the whole differentiation process (although with different duration, depending on the studies), a general consensus exists concerning the decrease in gene expression of *Pparγ,* the master regulator of adipogenesis (Figure 3). In the single study addressing the phases of the adipogenic process in which isorhamnetin could exert its effects [19], the authors concluded that this molecule affected just the latest stages of pre-adipocytes differentiation, because it was only when cells were incubated on day three or six, but not on day zero, that they showed a reduction in triglyceride accumulation. It was also demonstrated that the main anti-adipogenic effect was a reduction in *Pparγ* gene expression. 

Regarding human pre-adipocytes, a single study has been published [17]. Analogous to that observed in murine pre-adipocytes, isorhamnetin was proved to be effective at 10, 25 and 50 μM. Along with another study carried out in pre-adipocytes 3T3-L1, the Wnt pathway was revealed as a mediator of the anti-adipogenic effects of isorhamnetin (Figure 3). 

The effects of isorhamnetin on mature adipocytes have been scarcely studied to date; only two studies have been published, yielding contradictory results [16,22]. The reason justifying the discrepancy remains unclear. 

Isorhamnetin, conjugated with other molecules (isorhamnetin 3-*O*-β-d-glucopyranoside and isorhamnetin-3-*O*-d-glucuronide), has also been studied in other works [23,24]. It appears that these two derivatives are more powerful as anti-adipogenic molecules than isorhamnetin because, whereas they are able to reduce triglyceride accumulation in adipocytes at doses of 5 and 1 µM, isorhamnetin is only effective at 10 µM or higher doses.

## 4. Studies in Rodents

Studies using animal models with different experimental approaches aimed at analysing the potential anti-obesity effect of isorhamnetin and isorhamnetin glycosides have revealed beneficial effects on body weight management and energy metabolism. The vast majority of the studies have been carried out in rodent models (Table 3). Anti-obesity effects in humans have not yet been explored by interventional studies. Nevertheless, observational data, which did not have the same scientific evidence that clinical trials, show that daily isorhamnetin intake is lower in Polish adults showing central obesity than in healthy normal-weight subjects [33].

Among the studies addressed in rodents, only one study tested the effect of isorhamnetin on male mice. In the study reported by Ganbold et al. (2019), the effect of isorhamnetin in a non-alcoholic steatohepatitis (NASH) model induced by a carbon tetrachloride injection was analysed [21]. Seven-week-old male C57BL/6J mice were distributed into three experimental groups. The control group was fed a standard diet and the other two groups were fed a high-fat diet (60% of energy from fat, mainly from lard). Animals in one of these groups were orally administered 50 mg of isorhamnetin/kg body weight/day. The length of the experimental period was 25 days, but the phenolic compound was administered only from day 11 to day 24. After that period, as expected, mice from the control group showed a reduction in body weight gain, percentage of adipose tissue and lower caloric intake than mice fed with the high-fat diet. No differences in these parameters or adipocyte hypertrophy were found between the groups fed with the high-fat diet (supplemented or not with isorhamnetin). Regarding serum parameters, there were no statistical differences in triglyceride, total-cholesterol or HDL-cholesterol levels between both groups.

The effect of isorhamnetin in female mice, using a dietetic model of obesity, was studied by Zhang et al. (2016) [19]. In this study, seven-week-old female C57/BL6 mice were put on a chow (10% energy from fat) or a high-fat diet (60% of energy from fat, mainly from lard) for three months. Next, mice fed with the high-fat diet were divided into two experimental groups over for four weeks. The diet was supplemented with 1% of isorhamnetin for one of the groups. The high-fat diet feeding increased the body weight, and isorhamnetin was able to prevent this effect, along with an induced boost in the size of visceral adipocytes. It was further found that the reduction in body weight was not caused by either a decrease in food intake and triglyceride absorption or by an increase in body temperature. When looking at the mechanisms underlying these effects, the authors observed a depletion in gene expression of the fatty acid transporter *Cd36*, the regulators of the lipogenesis stearoyl-CoA desaturase 1 (*Scd1*) and in sterol regulatory element-binding protein 1c (*Srebp1c*) in adipose tissue. However, when compared to mice fed with the same but not-supplemented diet, no differences were observed in the gene expression of *Pparγ*, fatty acid transporter *Ap2* and *Lpl* in the rodents supplemented with isorhamnetin.

Isorhamnetin treatment also decreased fasting glucose and insulin levels, as well as the area under the curve on the glucose tolerance test. Moreover, insulin resistance was improved. The biochemical analysis also revealed that this molecule attenuated the diet-induced increase in leptin levels, indicating an improvement in leptin resistance. However, it was not able to significantly prevent the diet-induced decrease of adiponectin levels, although a trend towards higher levels was observed. As far as dyslipidemia is concerned, isorhamnetin significantly dwindled serum triglycerides, total-cholesterol and free fatty acids (FFA), but not LDL-cholesterol or HDL-cholesterol. The authors concluded that isorhamnetin was able to reduce body weight and ameliorate comorbidities associated with obesity.

Regarding genetic models of obesity, Zhang et al. (2016) studied the effect of isorhamnetin on 12-week-old female C57BL/KsJ-*ob/ob* mice [19]. The mice received (or did not) 100 mg of isorhamnetin/kg of body weight/day, orally, for two weeks. At the end of the experimental period, no differences in body weight or in food intake were observed between groups. When mRNA levels of several genes related to lipid metabolism were measured in the white adipose tissue, the authors observed that isorhamnetin significantly reduced *Pparγ*, *Ap2*, *Cd36*, *Lpl* and *Acc* gene expression, although not *FAS* expression. Regarding biochemical analysis, the supplementation improved glucose tolerance and reduced serum triglyceride levels. However, serum total-cholesterol, LDL-cholesterol and HDL-cholesterol were not modified by isorhamnetin.

Lastly, there are also studies devoted to analyzing the effects of isorhamnetin glycosides. In this regard, Rodríguez-Rodríguez et al. (2015) studied the potential anti-obesity effect of an *Opuntia fucus-indica* extract, containing more than 90% of the following isorhamnetin glycosides: isorhamnetinglucosyl-rhamnosyl-rhamnoside, isorhamnetin-glucosylrhamnosyl-pentoside, isorhamnetin-hexose-methylpentose-pentose, isorhamnetin-glucosyl-pentoside and isorhamnetin-glucosyl-rhamnoside [34] (Figure 4). For that purpose, male eight-week-old male C57BL/6NCrl mice were divided into four experimental groups. The control group was fed a standard diet containing 12% energy from fat and the other groups were fed a high-fat diet (45% energy from fat, mainly from lard) supplemented, or not, with 0.3 % or 0.6 % of the extract. The length of the experimental period was 12 weeks. 

As expected, a high-fat diet significantly increased body weight, and both doses of the extract completely prevented this effect. Mice treated with isorhamnetin glycosides showed no change in food intake, but they displayed a respiratory exchange ratio (RER) of 0.8 (32% oxidation of carbohydrates vs. 68% oxidation of fat), when a RER of 0.7 during the feeding period (0% oxidation of carbohydrates vs. 100% oxidation of fat) is already recognised as metabolic flexibility. Moreover, isorhamnetin increased energy expenditure. 

As far as adipose tissue morphology is concerned, mice supplemented with isorhamnetin glycosides had a reduction in adipocyte size in visceral and subcutaneous adipose tissue depots, which had a significant correlation with serum leptin content in all four groups. These results showed that the isorhamnetin glycosides prevent obesity, by reducing adipose tissue hypertrophy induced by the high-fat diet. 

Regarding serum parameters, the highest dose of the extract, albeit not the lowest one, decreased circulating total cholesterol, LDL-cholesterol and HDL-cholesterol in mice fed with high-fat diet, but did not modify plasma triglyceride levels. The extract also reduced serum glucose and insulin levels. HOMA index, an indicator of insulin resistance, was lessened by both doses of the extract, and in the case of the highest dose, the index value reached the level of the control group. Leptin, which regulates food intake and energy expenditure to maintain body fat stores, was decreased by the extract but not to the level of the control group.

### 4.1. Summary

To date, very few studies addressing rodents are available from which to draw clear conclusions concerning the anti-obesity effect of isorhamnetin. Among the three reported studies, only one has revealed an anti-obesity effect of this compound. Moreover, a study showed a significant reduction in body weight gain and adipocyte size when isorhamnetin glycosides were administered. It is important to emphasize that a short experimental period (two weeks) was applied for the two studies in which no significant effects were observed. In contrast, the animals were treated for longer periods (4–12 weeks) in the other two studies. Consequently, it seems that treatment periods longer than two weeks are needed by isorhamnetin and its glycosides in order to be effective as anti-obesity agents. Nevertheless, improvements in glycaemic control can be observed even in short treatments. Regarding the potential mechanisms of action involved in the anti-obesity of isorhamnetin, although some contradictory results are found between the two reported studies that address this issue, reductions in de novo lipogenesis and fatty acid uptake could be proposed. 

## 5. Studies in Other Animal Models

In addition to rodents, other animal models have been used to study the effect of isorhamnetin on obesity. *Caenorhabditis elegans* (*C. elegans*) is a small nematode in which human lipid metabolism pathways are well preserved. For that reason, it is considered a good model to evaluate potential obesity therapeutics. In this context, Farias-Pereira et al. (2019) studied the effects of isorhamnetin on lipid metabolism using this nematode [35]. Adult worms were treated or not with isoharmnetin (50, 100, 200 μM) for two days. At the end of the experimental period, animals treated with isorhamnetin showed lower triglyceride content, although the reduction was only significant at 100 and 200 μM. The mechanism of action was studied at both effective doses (100 and 200 μM). The lowering in fat accumulation was neither due to a decrease in food intake, nor to an increase in energy expenditure. Regarding lipogenesis, both doses increased the expression of *Cebp-2*, a transcription factor that regulates the expression of lipogenesis-related genes, including Fat-5, an orthologue of the human stearoyl-CoA desaturase. However, no changes in *Fat-5* mRNA levels were observed. These results indicate that, in spite of the change in the expression of the transcriptional factor, the lipogenesis was not likely involved in the delipidating effects in *C. elegans*. AMP-activated protein kinase (AMPK), which is encoded by AMPK subunits homologs *Aak-1* and *Aak-2* in *C. elegans,* is one of the major regulators of energy metabolism. Isorhamnetin treatment up-regulated both genes, but only at the highest dose, suggesting an AMPK activation. *Let-363* is a homolog of the human mechanistic target of rapamycin (mTOR) that coordinates systemic energy status. In this study only isorhamnetin at 100 μM up-regulated its gene expression, indicating that let-363/mTOR may not be involved in the effect of isorhamnetin.

Lipolysis is a metabolic pathway mainly regulated by adipose triglyceride lipase and hormone-sensitive lipase, homologs of *Atgl-1* and *Hosl-1,* respectively in *C. elegans.* In order to analyse the potential effect of isorhamnetin on this process, *Atgl-1* and *Hosl-1* gene expression was measured. The authors observed that *Atgl-1* was up-regulated by the highest dose of isorhamnetin and *Hosl-1* by the lowest one, hence suggesting the possible involvement of lipolysis in the delipidating effects. As far as β-oxidation is concerned, mRNA levels of *Nhr-49* (a homolog of PPARα)*, Mdt-15* (a homolog of mediator complex subunit 15)*, Acs-2* (a homolog of acyl-coA synthetase 2)*, Acs-11* (an ortholog of the acyl-CoA synthetase 3)*, Cpt-5* (an ortholog of the carnitine palmitoyl transferase 1)*, Ech-4* (an ortholog of the enoyl-CoA hydratase 2) and *Ech-1.1* (an ortholog of hydroxyacyl-CoA dehydrogenase trifunctional multienzyme complex) were analysed. No changes were observed in any of the genes, with the exception of the up-regulation of the *Nhr-49* and *Ech-1.1*, hintting that in fact, beta oxidation was not involved in the effects of isorhamnetin. In the case of *Ech-1.1*, the increase was only significant with at lowest dose. 

To confirm the potential mechanisms involved in the delipidating effects on *C. elegans,* the authors treated mutants of this worm with isorhamnetin at 100 and 200 μM. *C. elegans* mutant lacking Sterol regulatory element Binding Protein (Sbp-1), a homolog of SREBP, showed lower triglyceride content than the control animals. Thus, as previously presumed, lipogenesis was not involved in the fat-lowering effect of the phenolic compound. Ortholog of the mammalian TUBBY 1 (TUB-1) is associated with the regulation of lipid metabolism, probably via a neuroendocrine pathway in *C. elegans.* In that regard, isorhamnetin at 200 μM decreased triglyceride content in *tub-1* mutant by 46%, when compared with the controls. These results suggest that the effects of isorhamnetin on fat accumulation were independent in *Tub-1* in *C. elegans*. On the other hand, in *Aak-2* lacking animals treated with isorhamnetin 100 μM, triglyceride content was found to be lower than in control groups, and in *Aak-1* and *Aak-2* lacking animals both doses reduced its content. Taking into account that AMPK is encoded by the two subunits homologs *Aak-1* and *Aak-2* in *C. elegans,* it can be suggested that probably AMPK is not involved in this effect at transcriptional level. However, it is possible AMPK involvement at a post-transcriptional level. 

In this nematode, nuclear hormone receptor NHR-49 (a homolog of the peroxisome proliferator-activated receptor α [PPARα]), regulates fatty acid β-oxidation and lipolysis. The reduced fat accumulation induced by isorhamnetin was abolished in *nhr-49* mutant, suggesting that isorhamnetin fat-lowering effects were dependent on *nhr-49*. Also in these mutants, isorhamnetin at 100 μM and 200 μM up-regulated the transcription of *nhr*-49 by 19% and 20%, respectively. Isorhamnetin neither changed the expression of a mediator complex that interacts with NHR-49 protein, *mdt-15* (a homolog of the human mediator complex subunit 15, MED15), nor the NHR-49-downstream genes: *acs-2* (a homolog of the human acyl-coA synthetase 2), *acs-11* (an ortholog of the human acyl-CoA synthetase 3), *cpt-5* (an ortholog of the human carnitine palmitoyl transferase 1, CPT1) and *ech-4* (an ortholog of the human enoyl-CoA hydratase 2). Nevertheless, 100 μM isorhamnetin up-regulated the expression of *ech-1.1* (an enoyl-CoA hydratase, an ortholog of the human hydroxyacyl-CoA dehydrogenase trifunctional multienzyme complex, HADHA) by 200% compared to the control. Effects of isorhamnetin on *ech-1.1* expression were dependent to *nhr*-49 by using *nhr-49* mutant, and the up-regulation of *ech-1.1* by isorhamnetin was abolished in *nhr-49* mutant. This suggests that isorhamnetin increases expression of *ech-1.1* via NHR-49-dependent pathway. In conclusion, the reduction of fat accumulation by isorhamnetin in *C. elegans* depends on fatty acid β-oxidation, lipolysis and AMPK activation. The results obtained with the performed experiments with mutant nematodes are summarized in Table 4. 

## 6. Concluding Remarks

The reported data presented in this review show that, in vitro, isorhamnetin is able to reduce adipogenesis. Among the doses used, only 10 μM and higher were effective. Taking into account that these doses are greater than the concentrations found in plasma from treated animals, it is not evident that the effects observed in these studies can be reproduced in vivo [36]. 

Regarding the studies carried out in animal models, it can be stated that isorhamnetin and its glycosides show anti-obesity actions, for they limit body fat accretion, further ameliorating glycaemic control and, in general terms, serum lipid profile (Figure 5). However, it seems that, in order to observe these positive effects, periods of treatments longer than two weeks, in the case of rodents, are needed. In nematodes, it has been observed that their anti-obesity effects are probably via lipolysis, b-oxidation and AMPK activation.

Due to the fact that only a reduced number of studies addressing the anti-obesity effects of isorhamnetin and its glycosides have been reported so far, further research is needed to increase the scientific evidence concerning this issue. In addition, studies in animal models are also essential to establish the mechanisms of action underlying this effect. Finally, well designed randomised clinical trials are required to know whether the positive results observed in animals may also be found in humans, to further determine if isorhamnetin and its glycosides can indeed represent a real tool to prevent and/or treat body fat excess. 

## Figures and Tables

**Figure 1 ijms-24-00299-f001:**
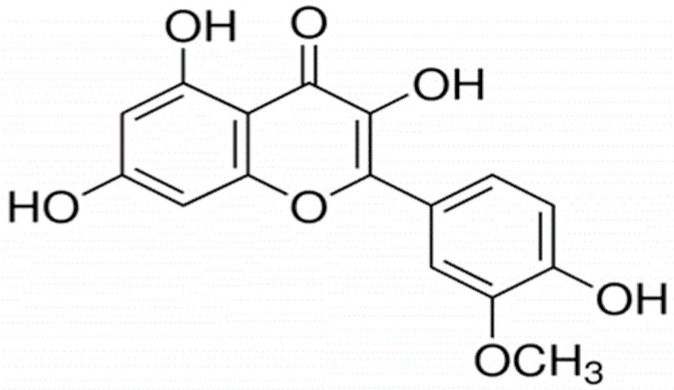
Chemical structure of isorhamnetin.

**Figure 2 ijms-24-00299-f002:**
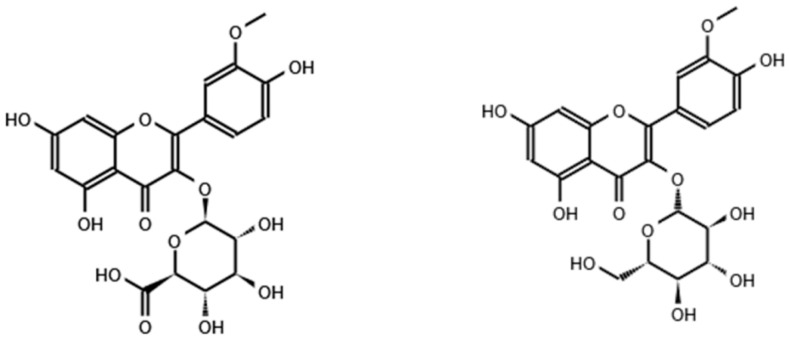
Chemical structure of isorhamnetin 3-*O*-d-glucuronide (**left**) and isorhamnetin 3-*O*-β-glucopyranoside (**right**).

**Figure 3 ijms-24-00299-f003:**
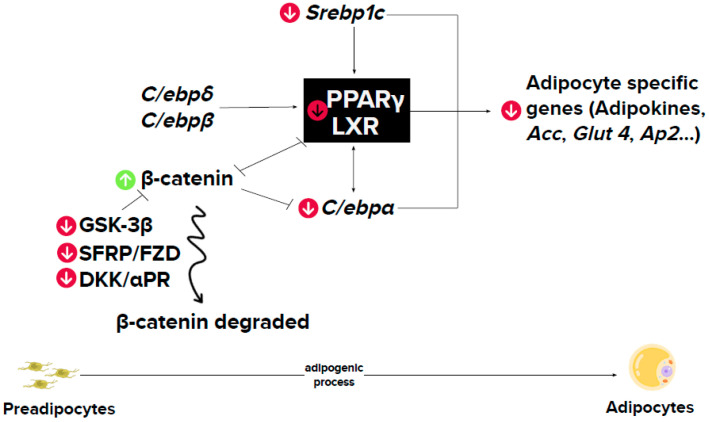
Mechanism of action underlying the effect of isorhamnetin on adipogenesis. αPR: progesterone receptor α; ACC: acetyl coA carboxylase; AP2: adipose fatty acid-binding protein 2; C/ebp: CCAAT-enhancer-binding protein; DKK: Dickkopf; FZD: frizzled class receptor; GLUT4: glucose transporter member 4; GSK-3β: glycogen synthase kinase-3 beta; PPARγ: peroxisome proliferator activated receptor γ; RXRα: retinoid X receptor alpha; SFRP: secreted frizzled related protein; SREBP1c: sterol regulatory element-binding protein 1c. Green arrows: upregulation by isorhamnetin; red arrows: downregulation by isorhamnetin.

**Figure 4 ijms-24-00299-f004:**
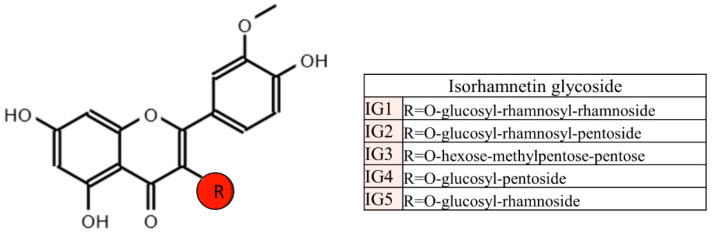
Isorhamnetin glycosides found in Opuntia-ficus indica.

**Figure 5 ijms-24-00299-f005:**
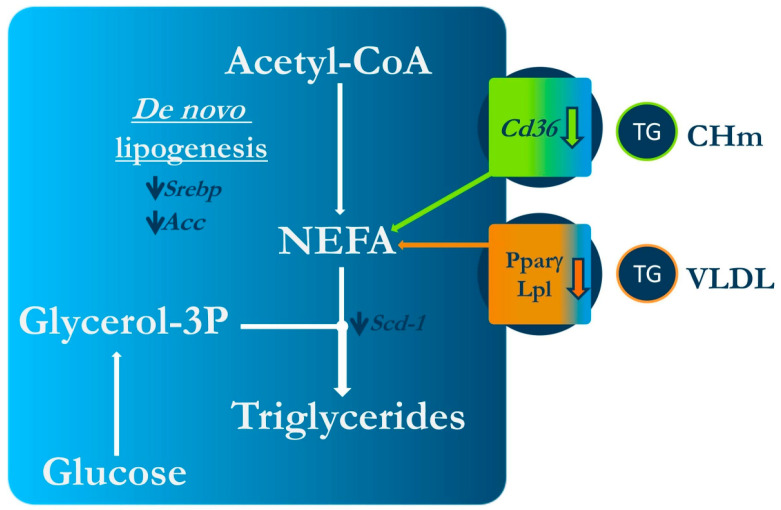
Mechanism of action underlying the effect of isorhamnetin on obesity in rodents. Acc: acetyl-CoA carboxylase; Cd36: cluster of differentiation 36; CHm; Chylomicrons; Lpl: lipoprotein lipase; NEFA: non-sterified fatty acids; PPARγ: peroxisomal proliferator-activated receptor γ; Scd-1: stearoyl-CoA desaturase-1; Srebp: sterol regulatory element binding protein; VLDL: very light density lipoproteins.

**Table 1 ijms-24-00299-t001:** Content of isorhamnetin and some of its glycosides in different foods and plants.

Food Source or Plant	Mean Content	SD	Ref.
**Isorhamnetin**			
Yellow onion	9.31 mg/100 g	21.65	[5]
Red onion	1.51 mg/100 g	2.37	[5]
Red wine	0.33 mg/100 mL	0.14	[5]
Almond	0.14 mg/100 g	0.13	[5]
Sea-buckthornberry juice	0.09 mg/100 mL	0.07	[5]
Ginkgo biloba leaves (seasonal variation)	23–693 mg/100 g	-	[6]
**Isorhamnetin 3-*O*-glucoside**			
Pear	0.45 mg/100 g	0.42	[5]
Green grapes	0.17 mg/100 g	0.19	[5]
Apple	NQ		[7]
Wild strawberry	1.14 mg/100 mL	0.12	[8]
**Isorhamnetin 3-*O*-galactoside**			
Almonds	0.58 mg/100 g	0.19	[5]
Apple	NQ		[7]
Wild strawberry	0.44 mg/100 mL	0.05	[8]
Wild blackberry	1.25 mg/100 mL	0.1	[8]
**Isorhamnetin 3-*O*-glucoside 7-*O*-rhamnoside**			
Sea-buckthornberry juice	7.05 mg/100 mL	0.64	[5]
**Isorhamnetin glucoxyl-rhamnosyl-pentoside**			
Prickly pear from *Opuntia stricta* var. *dilleni*	5.01 mg/100 g	0.0	[2]
**Isorhamnetin glucoxyl-rhamnosyl-rhamnoside**			
Prickly pear from *Opuntia dillenii*	0.4 mg/100 g	0.02	[2]
**Isorhamnetin 3-*O*-hexoside**			
*Hippophae rhamnoides*	56.25 mg/100 g	0.1	[9]
**Isorhamnetin 3-*O*-hexoside-deoxyhexoside**			
*Hippophae rhamnoides*	60.82 mg/100 g	0.6	[9]
Sweet cherries	0.09 mg/100 g	0.03	[10]
Sour cherries	1.57 mg/100 g	0.48	[10]

NQ: non-quantified; SD: standard deviation. Ref.: reference.

**Table 2 ijms-24-00299-t002:** Effects of isorhamnetin in cultured cells.

Reference	Compound and Dose	Cell Type and Experimental Design	Effects	Mechanisms of Action
Isorhamnetin				
Lee et al. (2009) [16]	Isorhamnetin 1, 10, 25, 50 μM 25, 50 μM	3T3-L1 pre-adipocytes The 9 days of the differentiation period Mature adipocytes Treatment of 20 h on day 9 after differentiation	↓ TG content ↓ Adipocyte differentiation (25 and 50 μM) NS TG content	↓ *Pparγ*, *C/ebpα*, *Pgc-1*, *Lpl*, *Cd36*, *Ap2* and *Lxr-α* gene expression NS *C/ebpβ*, *C/ebpδ* and *Krox* gene expression ↓ GPDH activity ↓ Adiponectin gene and protein expressions ↓ *Il-6, Mcp*-1 and *Pai-1* gene expressions NS *Pparγ* gene expression
Lee et al. (2010) [17]	Isorhamnetin 1, 10, 25, 50 μM	Human mesenchymal stem cells The 21 days of the differentiation period	↓ Adipocyte differentiation (10, 25 and 50 μM)	↓ *Dkk1,* S*frp1*, *C/ebpα*, *Pparγ* gene expression NS S*frp2*, S*frp3*, S*frp4* and *Dkk3* gene expression ↓ *Lrp5*, *Lrp6*, *Fzd4*, *Fzd6* and *Fzd7* gene expression ↑ Total and nuclear β-catenin protein expression ↑ GSK 3β phosphorylation
Lee et al. (2011) [18]	Isorhamnetin 50 μM	3T3-L1 pre-adipocytes The 6 days of the differentiation period	↓ TG content ↓ Adipocyte differentiation	↑ Nuclear β-catenin protein expression ↑Wnt/β-catenin signalling activity
Zhang et al. (2016) [19]	Isorhamnetin 10, 25 and 50 μM 50 μM	3T3-L1 pre-adipocytes The 9 days of the differentiation period Treatment at different stages; from day 0, 3 or 6 of differentiation	↓ TG content ↓Adipocyte differentiation (all doses) ↓ TG content (treatment at day 3 or later)	Treatment at day 0: *↓ C/ebpα*, *Pparγ* and *Ap2* gene expression NS *Pparγ* and *C/ebpβ* *Cd36*, *Acc*, *Ucp2*, *Lpl* gene expression Treatment at day 6: *↓ Pparγ*, *Ap2*, *Cd36*, *Fas*, *Acc*, *Ucp2* and *Lpl* gene expression
Lee and Kim (2018) [20]	Isorhamnetin 0.1, 0.5, 1, 10, 20, 50 μM for cell viability 0.1, 0.5, 1, 10, 20 μM for TG accumulation	3T3-L1 pre-adipocytes The 7 days of the differentiation period	↓ TG content ↓ Adipocyte differentiation (20 μM)	↓ GPDH activity ↓ *Pparγ* and *Ap2* gene expression
Ganbold et al. (2019) [21]	Isorhamnetin 10 and 20 μM	3T3-L1 pre-adipocytes The 14 days of the differentiation period	↓ TG content (20 μM)	Not analysed
Eseberri et al. (2019) [22]	Isorhamnetin 0.1, 1 and 10 μM	3T3-L1 pre-adipocytes The 8 days of the differentiation period Mature adipocytes treated on day 12 after differentiation for 24 h	↓ TG content ↓ Adipocyte differentiation (10 μM) NS TG content	NS *C/ebpβ*, *Srebf1*, *C/ebpα*, *Dgat1*, *Dgat2* ↓ *Bcl2* gene expression ↑ *Trp3* gene expression NS *Cas3* gene expression
Conjugated Isorhamnetin				
Kong and Seo (2012) [23]	Isorhamnetin 3-*O*-β-d-glucopyranoside 1, 10 and 20 μM	3T3-L1 pre-adipocytes The 6 days of the differentiation period	↓ TG content ↓ Adipocyte differentiation (all doses)	↓ PPARγ, SREBP1, C/EBPα protein expression ↓ FAS, GLUT4, RXRα and leptin protein expression ↑ AMPK activity
Im et al. (2017) [24]	Isorhamnetin-3-*O*-d- glucuronide 5, 10 and 20 μM	3T3-L1 pre-adipocytes From day 4 to day 8 after differentiation	↓ TG content	Not analysed

ACC: acetyl coenzyme A carboxylase; AMPK: AMP-activated protein kinase; AP2: adipose fatty acid-binding protein 2; BCL2: cell leukaemia/lymphoma 2; CAS3: caspase 3; CD36: cluster of differentiation 36; C/EBP: CCAAT-enhancer-binding protein; CPT-1α: carnitine palmitoyl transferase-1α; DGAT: diacylglycerol O-acyltransferase; DKK: Dickkopf; FAS: fatty acid synthase; FZD: frizzled class receptor; GLUT4: glucose transporter member 4, GPDH: glycerol-3-phosphate dehydrogenase; GSK-3β: glycogen synthase kinase-3 β; IL-6: interleukin-6; LPL: lipoprotein lipase; LRP: LDL Receptor Related Protein; LXR-α: liver X receptor α; MCP-1 monocyte chemoattractant protein-1; NS: not significant; PAI-1; plasminogen activator inhibitor-1; PGC-1: peroxisome proliferator-activated reporter gamma coactivator-1; PPARγ: peroxisome proliferator activated receptor γ; RXRα: retinoid X receptor alpha; SFRP: secreted frizzled related protein; SREBF1: sterol regulatory element-binding factor 1; SREBP1: sterol regulatory element-binding protein 1; TFAM: mitochondrial transcription factor A; TG: triglycerides; TRP53: transformation related protein 53; UCP2: uncoupling protein 2; WNT: wingless-type mammary tumour; ↓: decrease; ↑: increase.

**Table 3 ijms-24-00299-t003:** Effects of isorhamnetin on body weight, adipose tissue and serum parameters in rodents.

Reference	Compound	Experimental Design	Effects	Mechanisms of Action
Isorhamnetin				
Zhang et al. (2016) [19]	Isorhamnetin 1% in the diet Isorhamnetin 100 mg/kg BW/day by gavage	Female 7-week-old C57/BL6 mice Standard or high-fat diet (60% energy from fat) Experimental period length: 4 weeks Female 12-week-old C57BL/KsJ-*ob/ob* mice Standard diet Experimental period length: 2 weeks	↓ Body weight ↓ Size of visceral adipocytes ↓ Fasting serum glucose and insulin levels and insulin resistance ↓ Leptin resistance ↓ Serum TG, total-cholesterol and FFA NS LDL- and HDL-cholesterol NS Body weight ↑ Glucose tolerance ↓ Serum TG NS serum total-, LDL- and HDL-cholesterol	NS *Pparγ*, *Ap2* and *Lpl* gene expression ↓ *Cd36*, *Scd1* and *Srebp1c* gene expression NS *FAS* gene expression ↓ *Pparγ*, *Ap2 Cd36*, *Lpl, Acc* gene expression
Ganbold et al. (2019) [21]	Isorhamnetin 50 mg/kg BW/day	Male 7-week-old C57BL/6J mice with NASH Standard or high-fat diet (60% of energy from fat) Experimental period length: 2 weeks	NS body weight, adipose tissue and adipocyte size NS serum total and HDL-cholesterol	
Isorhamnetin glycosides				
Rodríguez-Rodríguez et al. (2015) [34]	*Opuntia fucus-indica* extract with isorhamnetin glycosides 0.3% or 0.6% in the diet	Male 8-week-old C57BL/6NCrl mice Standard or high-fat diet (45% of energy from fat) Experimental period length: 12 weeks	↓ Body weight ↑ Energy expenditure ↓ Adipocyte size in visceral and subcutaneous adipose tissues ↓ Serum total cholesterol, LDL-cholesterol and HDL-cholesterol (0.6%) NS Serum TG ↓ Serum glucose and insulin levels and HOMA-IR ↓ Serum leptin	

ACC: acetyl coenzyme A carboxylase; AP2: adipose fatty acid-binding protein 2; BW: body weight; CD36: cluster of Differentiation 36; FAS: fatty acid synthase; FFA: free fatty acids; HOMA-IR: insulin resistance homeostatic model assessment; LPL: lipoprotein lipase; NASH: non-alcoholic steatohepatitis; NS: not significant changes; PPARγ: peroxisome proliferator activated receptor γ; SCD1: stearoyl-CoA desaturase 1; SREBP1c: sterol regulatory element-binding protein 1c; TG: triglyceride; UCP2: uncoupling protein 2. ↓: decrease; ↑: increase.

**Table 4 ijms-24-00299-t004:** Effects of isorhamnetin treatment in mutant nematodes lacking lipid metabolism related genes.

Lacking Gene	Related Alteration	Effect
Sbp-1	Lipogenesis	↓TG
Tub-1	Lipogenesis	↓TG
Nhr-49	b-oxidation, lipolysis	NS
Aak-2	Energy metabolism	↓TG
Aak-1 and Aak-2	Energy metabolism	↓TG

Aak: AMP-activated kinase; Nhr-49: nuclear hormone receptor 49; Sbp-1: sterol regulatory element binding protein 1; TG: triglyceride; Tub-1: ortholog of the mammalian TUBBY 1.

## Data Availability

No applicable.

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
