# Peer review of "Anti-Obesity Effects of Isorhamnetin and Isorhamnetin Conjugates"

_ijms, 2022, doi:10.3390/ijms24010299_

Round 1

Reviewer 1 Report

Comments for authors:

1- I think the title should be modified with the addition of " isorhamnetin conjugates or derivatives" since you also review their anti-obesity effects in the article.

2- Table 1: why you only mentioned the food sources of isorhamnetin, It is better to add other plant sources already mentioned in the introduction such as Opuntia, Gingiko, and other important medicinal plants.

3- check the italic form of 'in vitro' and 'in vivo' in the whole article.

4- Why do you focus most on the 3T3-L1 cell line, there are no studies with other types of cells.?

5- Do you know why the clinical trials are not yet done with isorhamnetin, do you think is it for toxicity reasons? 

6- Figure 3:  Reconsider it. It is better to make a figure summarizing the mechanisms of action described in section 2.

Author Response

Comments for authors:

1- I think the title should be modified with the addition of "isorhamnetin conjugates or derivatives" since you also review their anti-obesity effects in the article.

We acknowledge the reviewer for this comment. The new title is "Anti-obesity effects of isorhamnetin and isorhamnetin conjugates", that, as indicated by the reviewer, reflects better the content of this review.

2- Table 1: why you only mentioned the food sources of isorhamnetin, It is better to add other plant sources already mentioned in the introduction such as Opuntia, Gingiko, and other important medicinal plants.

According to reviewer's comment, the Table 1 has been modified.

3- Check the italic form of 'in vitro' and 'in vivo' in the whole article.

According to reviewer's suggestion, both in vitro and in vivo word have been checked and all of them are written in italics, with the exception of the titles of the manuscripts in the section"References" (respecting the reference editor system used).

4- Why do you focus most on the 3T3-L1 cell line, there are no studies with other types of cells?

Effectively, only studies carried out in 3T3-L1 adipocytes have been included because studies using other types of adipocyte cells have not been published so far, with the exception of a study addressed in human mesenchymal stem cells , that was already included in the previous version (Lee et al., 2010; ref 23).

5- Do you know why the clinical trials are not yet done with isorhamnetin, do you think is it for toxicity reasons? 

We have checked the literature and the information that we have obtained indicates that isorhamnetin is not a toxic compound (Li et al., 2022 doi: 10.2174/1381612828666220829113132). Thus, we believe that likely, the main reason is that the number of studies carried out in cell cultures and experimental animals is still scarce.

6- Figure 3: Reconsider it. It is better to make a figure summarizing the mechanisms of action described in section 2.

Following the reviewer's suggestion, a new Figure explained the mechanisms of action described in rodents has been included (Figure 5).

Reviewer 2 Report

The article by González-Arceo et al. is a valuable review of information about the potential impact of isorhamnetin on obesity. The Authors analyzed the data from in vitro and animal models studies, however they did not include any information about the available observations from humans. I suggest to add the information from human studies to fully present the topic.

The manuscript is well organized, with clear tables and figures. The language quality is satisfactory. In general, the manuscript has a high potential, but in many spots, it lacks relevant references for the stated information. 

The authors also did not provide the information about the search strategy used for this review. I suggest to add this information.

Line 40 – The authors mention that the main sources are: cherries, apples, pears and blackberries. Then the information in table 1 is given for e.g. onions, without the mentioned foods. Please be consistent and provide additional information in the table 1 or add information in line 40. 

Line 43 – Please give examples of species.

Line 55 – Please provide more relevant up-to-date (max. 3 years) citations of such observations (CVD, cancers, DM, obesity) in vitro, in animal models and human studies.

Line 265 – Please provide references to these two studies.

Line 268-269 – Please provide references for “other works”.

Line 278 – “Anti-obesity effects in humans have not yet been explored”. There are no interventional human studies regarding relationship between isorhamnetin and obesity available, however there are few human observational studies already published upon this topic (10.1017/S0007114520004754 10.3390/nu14235051 10.3390/nu10080991 ).

Author Response

The article by González-Arceo et al. is a valuable review of information about the potential impact of isorhamnetin on obesity. The Authors analyzed the data from in vitro and animal models studies, however they did not include any information about the available observations from humans. I suggest to add the information from human studies to fully present the topic.

The manuscript is well organized, with clear tables and figures. The language quality is satisfactory. In general, the manuscript has a high potential, but in many spots, it lacks relevant references for the stated information. 

The authors also did not provide the information about the search strategy used for this review. I suggest to add this information.

We acknowledge the reviewer for this comment. A new methodological section has been included in this revised version (lines 71-74).

Line 40 – The authors mention that the main sources are: cherries, apples, pears and blackberries. Then the information in table 1 is given for e.g. onions, without the mentioned foods. Please be consistent and provide additional information in the table 1 or add information in line 40. 

According to the reviewer's comment, Table 1 has been completed. In addition, other sources previously included in the Table 1 (first version) have been included in the text (lines 41-44).

Line 43 – Please give examples of species.

According to reviewer's comment, two examples have been provided.

Line 55 – Please provide more relevant up-to-date (max. 3 years) citations of such observations (CVD, cancers, DM, obesity) in vitro, in animal models and human studies.

According to reviewer's comment, more actual references have been included.

Line 265 – Please provide references to these two studies.

Following the reviewer's suggestion, new references have been included.

Line 268-269 – Please provide references for “other works”.

Following the reviewer's suggestion, new references have been included.

Line 278 – “Anti-obesity effects in humans have not yet been explored”. There are no interventional human studies regarding relationship between isorhamnetin and obesity available, however there are few human observational studies already published upon this topic (10.1017/S0007114520004754 10.3390/nu14235051 10.3390/nu10080991 ).

According to reviewers, the sentence has been modified in order to be more precise. The new sentence is “Anti-obesity effects in humans have not yet been explored in interventional studies”.

Concerning the papers proposed by the reviewer showing observational studies, the study doi 10.1017/S0007114520004754 and the study doi: 10.3390/nu10080991 address the association between different flavoids (isorhamnetin among them) and metabolic syndrome, but no on obesity; thus we have not included them in the text because we consider that they do not provide precise information on the issue addressed in the present review. The study doi: 10.3390/nu14235051 shows significant differences in flavonol habitual intakes between participants with central obesity and healthy controls. This relationship was observed for total flavonols and for selected compounds, such as quercetin, kaempferol and isorhamnetin. Consequently, this study has been included in the text.

Round 2

Reviewer 2 Report

I would like to thank the Authors for the revised version of the manuscript. I believe that the corrections improved its quality, however I still have few minor comments, that have to be addressed before publication. 

Line 43 – please exchange the word „stuff” for “products”, or just use the term “foods” as the word stuff is too colloquial for this type of the article.

Please delete double spaces (line 44)

Line 56 - Nutrients formatting requires adding references in the text in the square brackets, not in with upper index. Please correct it throughout the whole manuscript.

Line 56 – the additional references are not added as declared by the authors. Please correct it.

Line 71 – please add the information about the search period – were there any searching time limits, when was the search performed.

Line 273 – the conclusions from the cited study are contrary to described by the authors. The isorhamnetin intake in the cited study was lower in patients with central obesity and higher in healthy ones. Please correct it, as the sentence is misleading.

Author Response

Line 43 – please exchange the word „stuff” for “products”, or just use the term “foods” as the word stuff is too colloquial for this type of the article.

According to reviewers, the word has been modified.

Line 56 - Nutrients formatting requires adding references in the text in the square brackets, not in with upper index. Please correct it throughout the whole manuscript.

We are sorry for the mistake, the reviewer is right and some citations in the text were not put in brackets. It has been corrected in the new version of the manuscript.

Line 56 – the additional references are not added as declared by the authors. Please correct it.

We apologise, because unintentionally we added the additional references in another paragraph of the text. This mistake is now corrected in the revised version.

Line 71 – please add the information about the search period – were there any searching time limits, when was the search performed.

This review is not a systematic one. For that reason, some of that information was not included. However, according to reviewers’ comment the data of the last search period has been added. Regarding the searching time limits, that information was included in the previous version.

Line 273 – the conclusions from the cited study are contrary to described by the authors. The isorhamnetin intake in the cited study was lower in patients with central obesity and higher in healthy ones. Please correct it, as the sentence is misleading.

We are sorry for the mistake, it has already been corrected